# Primary Fallopian Tube Carcinoma Presenting with a Massive Inguinal Tumor: A Case Report and Literature Review

**DOI:** 10.3390/medicina58050581

**Published:** 2022-04-23

**Authors:** Michihide Maeda, Tsuyoshi Hisa, Shinya Matsuzaki, Shuichi Ohe, Shigenori Nagata, Misooja Lee, Seiji Mabuchi, Shoji Kamiura

**Affiliations:** 1Department of Gynecology, Osaka International Cancer Institute, Osaka 541-8567, Japan; michihide.maeda@oici.jp (M.M.); hisa-tu@mc.pref.osaka.jp (T.H.); shinya.matsuzaki@oici.jp (S.M.); seiji.mabuchi@oici.jp (S.M.); 2Department of Dermatologic Oncology, Osaka International Cancer Institute, Osaka 541-8567, Japan; s.ohe@oici.jp; 3Department of Clinical Pathology, Osaka International Cancer Institute, Osaka 541-8567, Japan; shnagata@oici.jp; 4Department of Forensic Medicine, School of Medicine, Kindai University, Osaka 589-8511, Japan; misooja810@gmail.com

**Keywords:** primary fallopian tube carcinoma, inguinal tumor, inguinal lymph node metastasis, serous carcinoma

## Abstract

Primary fallopian tube carcinoma (PFTC) has characteristics similar to those of ovarian carcinoma. The typical course of PFTC metastasis includes peritoneal dissemination and pelvic and paraaortic lymph node metastasis, while inguinal lymph node metastasis is rare. Moreover, the initial presentation of PFTC with an inguinal tumor is extremely rare. A 77-year-old postmenopausal woman presented with a massive 12-cm inguinal subcutaneous tumor. After tumor resection, histopathological and immunohistochemical analysis showed that the tumor was a high-grade serous carcinoma of gynecological origin. Subsequent surgery for total hysterectomy with bilateral salpingo-oophorectomy revealed that the tumor developed in the fallopian tube. She received adjuvant chemotherapy with carboplatin and paclitaxel, followed by maintenance therapy with niraparib. There has been no recurrence or metastasis 9 months after the second surgery. We reviewed the literature for cases of PFTC and ovarian carcinoma that initially presented with an inguinal tumor. In compliance with the Preferred Reporting Items for Systematic Reviews guidelines, a systematic literature search was performed through 31 January 2022 using the PubMed and Google scholar databases and identified 14 cases. In half of them, it was difficult to identify the primary site using preoperative imaging modalities. Disease recurrence occurred in two cases; thus, the prognosis of this type of PFTC appears to be good.

## 1. Introduction

Primary fallopian tube carcinoma (PFTC) originates in the salpingeal mucosa and has similar dissemination patterns, clinical behavior, effects of debulking surgery, and chemosensitivity to those of ovarian carcinoma [1]. Its accurate incidence is unknown, as several studies have shown that ovarian and primary peritoneal carcinomas also originate in the fallopian tube [2,3,4,5]. Moreover, histological, molecular, and genetic analyses have shown that approximately 80% of high-grade serous carcinomas of the ovary or peritoneum may have originated at the fimbrial end of the fallopian tube [2,3]. The staging systems and treatment strategies used for PFTC, cytoreductive surgery, and adjuvant chemotherapy are the same as those used for epithelial ovarian carcinoma [1]. 

PFTC is often diagnosed at an advanced stage; 19%, 16%, 46%, and 19% of women are diagnosed at stages I, II, III, and IV, respectively [1,6]. PFTC cells disseminate primarily within the peritoneal or pleural cavity, with only superficial invasion of the intra-abdominal organs (e.g., omentum and small and large intestines). Therefore, PFTC often presents with abdominal distention, nausea, anorexia, abdominal pain, and venous thromboembolism [7,8,9,10], which are typically attributed to massive ascites, pleural effusion, and metastatic disease with omental or bowel involvement. However, palpable lymph nodes and metastasis to the umbilicus, breast, and brain have also been reported as rare initial symptoms of PFTC [11,12].

In this study, we report a rare case of PFTC that initially presented with a massive 12-cm inguinal subcutaneous tumor. We also review the literature for cases of PFTC with inguinal tumors as initial symptoms. After the tumor resection, histopathological and immunohistochemical (IHC) analysis suggested that the tumor was composed of metastatic lymph nodes from gynecological malignant disease, although no tumor was detected through contrast computed tomography (CT) and pelvic magnetic resonance imaging (MRI).

## 2. Case Presentation

A 77-year-old postmenopausal woman (gravida 2, para 2) was referred to our hospital due to right inguinal swelling. She had no past medical history or family history of malignant neoplasm. Examination revealed a 12 cm, hard, movable tumor in the right inguinal region. On plain MRI, the tumor showed a phyllodes-like multilocular cystic structure with an internal septum and a solid component. The solid component demonstrated high signal intensity on the diffusion-weighed image and low signal intensity on the apparent diffusion coefficient map (Figure 1). Serum squamous cell carcinoma antigen and CA19-9 were negative, and the serum CA125 level was 36 U/mL (reference ≤ 35 U/mL). The tumor was diagnosed as a subcutaneous soft-tissue malignant neoplasm and was resected by a dermatologist (Figure 2A).

Histopathological analysis revealed a poorly differentiated adenocarcinoma with extensive hemorrhage and necrosis (Figure 2B), with relatively small cuboidal carcinoma cells proliferating in a papillary fashion and a fibrous stroma. The multilocular mass was encapsulated and separated with fibrous septa, accompanied by a marginal lymph node also involved by metastasis. Immunohistochemical (IHC) staining was positive for PAX-8 (Figure 2E), focally positive for WT-1 (Figure 2F), showed overexpression of p53 (Figure 2G), and was positive for cytokeratin (CK) 7 (Figure 2D), but negative for CK20, CDX-2, and GATA3. Based on the results of the histopathological and IHC analyses, the tumor was diagnosed as a metastasis of gynecological origin; hence, the patient was referred to our department.

Transvaginal ultrasonography, contrast-enhanced pelvic MRI, and thoracoabdominal-pelvic computed tomography (CT) (Figure 3A,B) revealed hydrometra and no other abnormal findings in the uterus or ovaries. Uterine cervical cytology was negative for intraepithelial lesions. For histopathological analysis, total hysterectomy was performed with bilateral salpingo-oophorectomy and omentectomy. There were no specific intraoperative findings, except for mild edema of the left fallopian tube (Figure 3C). Peritoneal washing cytology revealed the presence of adenocarcinoma cells. The surgery was performed without complications.

Histopathological analysis of the fallopian tube revealed the primary tumor defined to the left fimbria with exposure to the surface (Figure 3D). Papillary adenocarcinoma with a destructive solid pattern indicated stromal invasion (Figure 3E). Considering all the results, the patient was diagnosed with fallopian tube high-grade serous carcinoma with a massive inguinal lymph node metastasis. Based on the 2014 International Federation of Gynecology and Obstetrics (FIGO) and tumor, node, metastasis (TNM) staging systems for ovarian carcinoma, the postoperative surgical staging was pT1aNxM1 stage IVB [13,14].

Homologous recombination deficient status and breast cancer 1 and 2 gene (BRCA1/2) mutations were examined by the Myriad my Choice HRD™ test (Myriad Genetics Laboratories, Salt Lake City, UT, USA). The patient’s tumor showed homologous recombination repair deficiency, but no BRCA1/2 mutation was detected. She refused bevacizumab therapy due to the concern of intestinal perforation. She received adjuvant chemotherapy with carboplatin (AUC 5) and paclitaxel (175 mg/m^2^) every 3 weeks for six cycles, as well as maintenance therapy with niraparib (200 mg) once a day. There has been no recurrence or metastasis 9 months after the second surgery.

## 3. Discussion

The present case study of PFTC demonstrates that (1) inguinal swelling, although rare, can be an initial symptom; (2) the metastatic tumor can be considerably larger than the primary tumor; and (3) surgical resection of the adnexa is essential when IHC indicates gynecological origin of an inguinal node metastasis, even in the absence of abnormal imaging findings.

Although PFTC is a rare gynecologic malignancy, the incidence of the serous type has increased 4.19-fold from 2001 to 2014 [15]. The reason for this increasing trend is debatable, but it may be partly attributed to the greater recognition of PFTC by pathologists. Serous tubal intraepithelial carcinoma lesions were first reported in 2001 in tubal specimens from women with BRCA1/2 gene mutations [16,17]. Over the following decade, there was mounting histopathological evidence on the role of these lesions in the occurrence of high-grade serous carcinoma. Moreover, epidemiological studies have suggested that tubal ligation has a protective effect against ovarian carcinoma. Kurman and Shih proposed a theory on the pathogenesis of ovarian carcinoma, suggesting that it begins in the fallopian tube, which has been widely accepted [4]. Numerous retrospective studies have since shown a substantial reduction in the incidence of ovarian carcinoma after tubal ligation or opportunistic salpingectomy [18,19,20]. 

In the present case, the PFTC initially presented with a massive inguinal subcutaneous tumor, which was identified as a metastatic tumor of gynecological origin, although no apparent primary lesion was detected radiologically. Postoperative histopathological analysis showed that the tumor had originated from the fallopian tube. Hence, we concluded that the subcutaneous inguinal tumor was a rare initial symptom of PFTC.

Next, we reviewed the literature for similar cases. According to the 2020 edition of the Preferred Reporting Items for Systematic Reviews and Meta-Analyses statement [21], we conducted a systematic literature search in the PubMed and Google scholar databases (first 100 hits) from their inception to 31 January 2022, as previously performed with modification [22,23,24]. The keywords “inguinal lymph node” or “groin lymph node” and “fallopian tube carcinoma,” “ovarian carcinoma,” or “primary peritoneal carcinoma” were used to search. The search was limited to English literature, and only studies of PFTC with inguinal tumors as an initial symptom were included in the review. The exclusion criteria were as follows: (1) the tumor was diagnosed as inguinal hernia; (2) lack of information; (3) lack of histological analysis of the ovaries or fallopian tubes; and (4) conference papers, review articles, and systematic reviews.

We identified 13 studies with 13 eligible cases [25,26,27,28,29,30,31,32,33,34,35,36,37]. All 14 cases, including ours, are summarized in Table 1. The primary site was not identified in half of the cases (7/14), in one case not even by PET-CT. In the remaining cases, the primary site was confirmed by PET-CT (3/7), CT (3/7), or transvaginal ultrasonography (1/7). The median size of the inguinal lymph nodes was 3 cm (1.9–12 cm). In the majority of cases (10/14), the malignant disease was limited to the adnexa, while abdominal dissemination and pelvic lymph node metastasis were observed in three and one cases, respectively. These findings suggest that the primary sites of ovarian or fallopian tube carcinomas that initially present with an inguinal tumor are relatively small and not easily detected on imaging.

Primary reduction surgery was performed in 12 of the 14 cases [27,28,29,30,31,32,33,34,35,36,37], and, in two cases, neoadjuvant chemotherapy was administered, followed by interval debulking surgery [25,26]. Pelvic and paraaortic lymph node dissection was performed in seven cases, and lymph node metastasis other than inguinal was confirmed only in two cases. With regard to surgery, complete resection was performed in 11, optimal in 2, and suboptimal in 1 case. These results suggest that most women with ovarian carcinoma or PFTC (10/14 (71.4%)) with initial symptoms of inguinal tumors had limited adnexal disease with isolated inguinal lymph node metastasis.

Among the women who underwent complete resection (*n* = 11), no recurrence was detected during the follow-up period. Of the remaining three women, two women who had multiple metastases at the time of debulking surgery experienced disease recurrence 8 and 13 months after surgery, respectively. According to these findings, although inguinal lymph node metastasis is classified as stage IV ovarian or fallopian tube carcinoma, patients with ovarian or fallopian tube carcinoma initially presenting with an inguinal tumor appear to have a good prognosis. 

A population-based study examined the prognosis of women with inguinal lymph node metastases using the Surveillance, Epidemiology, and End Results database [38]. Women with stage IV ovarian carcinoma due to positive inguinal nodes were found to have better prognosis (5-year overall survival 46.3%) than those of the other groups (stage IV with positive distant nodes (32.9%, *p* < 0.001), and stage IV with distant metastases (25.3%, *p* < 0.001)) [38]. However, there was no available information regarding the initial symptoms of each patient. Therefore, the prognosis of patients with fallopian tube carcinoma initially presenting with an inguinal tumor was not examined.

A previous study examined the lymphatic drainage pathways in the ovaries of female fetuses after miscarriage or abortion [39]. This study revealed that there were three lymphatic drainage pathways of the ovaries: the abdominal and pelvic pathways were the major, and the inguinal was a minor pathway. While the detection of sentinel lymph nodes in ovarian carcinoma is difficult, the theory that the abdominal and pelvic lymphatic pathways are major pathways of lymph node metastasis is robust [40].

The other hypothesis regarding PFTC initially presenting with inguinal lymph node metastasis is a synchronous high-grade serous carcinoma in the fallopian tube and the canal of Nuck. The canal of Nuck is an abnormal patent pouch of peritoneum caused by the failure of closing processus vaginalis [41]. It creates a pathway between the peritoneal cavity and the female inguinal [42] and, for instance, inguinal endometriosis occurs via the canals of Nuck [43]. Therefore, there is a possibility that our case is a synchronous primary cancer in fallopian tube carcinoma and ectopic primary peritoneal cancer arising from inguinal endometriosis. Nevertheless, fallopian tube carcinoma with inguinal lymph node metastasis was reasonable, as the histopathological findings are similar between the cancer in the fallopian tube and inguinal lymph node (Figure 2C).

While there are some possible hypotheses regarding PFTC initially presenting with inguinal lymph node metastasis, we believe that a large inguinal lymph node metastasis with a small primary tumor is a unique type of ovarian and fallopian tube carcinoma. Some authors considered that the immune system may play a key role in this unique type of carcinoma, but the precise mechanism is unknown; thus, further molecular research is warranted to reveal the characteristics and mechanism of metastasis of this unique type of carcinoma [32,44].

The primary sites of inguinal lymph node metastasis of unknown origin are the skin of the lower extremities, cervix, vulva, trunk, rectum, anus, ovary, and penis [44]. Owing to its rarity, the frequency of PFTC metastasis to the inguinal lymph nodes is unknown. In our case, IHC staining facilitated the differential diagnosis of metastatic inguinal lymph nodes. The IHC staining pattern of high-grade serous PFTC was consistent with that of ovarian high-grade serous carcinoma [45]: often positive for cytokeratin 7 (CK 7), WT-1, PAX-8, and estrogen receptor, and negative for CK 20 and CDX-2.

The strength of this study is that it is likely the first to review the prognosis of ovarian and fallopian tube carcinoma with an initial symptom of inguinal tumor. We believe that our case presentation is also useful for clinicians to understand the rare initial symptoms of ovarian and fallopian tube carcinomas. Nevertheless, this study has several limitations. First, the results of our review may have a publication bias and require careful interpretation. For instance, the poor prognosis of ovarian and fallopian tube carcinomas with an initial symptom of inguinal tumor may not be reported. 

Second, since all studies included in this review were case reports, a severe bias may exist. Third, pelvic and paraaortic lymphadenectomy were not performed in our case, as a previous randomized trial that examined the role of lymphadenectomy in women with advanced ovarian carcinoma did not show a survival benefit [46]. As pelvic and paraaortic lymph node metastases were not determined, it remains unknown whether the inguinal lymph node metastasis was isolated. Fourth, we believe that the characteristics of our case were unique and molecular examination was not performed. To prove that our hypothesis is appropriate, molecular examinations, such as IHC and genetic analyses, are needed.

## 4. Conclusions

In women with unclear primary tumors with inguinal lymph node metastasis, diagnostic laparotomy or advanced imaging examination, such as PET-CT, may be recommended when the IHC analysis suggests malignancy of gynecological origin.

## Figures and Tables

**Figure 1 medicina-58-00581-f001:**
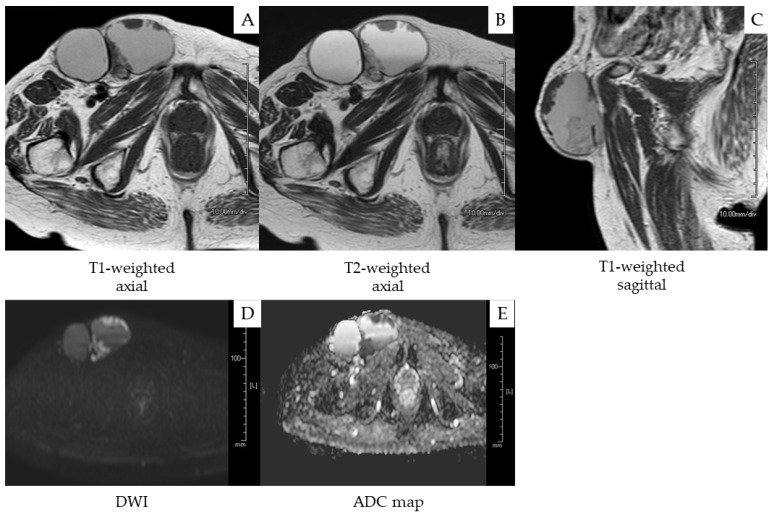
Magnetic resonance images of the inguinal tumor. (**A**) T1-weighted axial, (**B**) T2-weighted axial, and (**C**) T1-weighted sagittal images showing a phyllodes-like multilocular cystic tumor with an internal septum and a solid component. The solid component exhibits (**D**) high signal intensity on the diffusion-weighted image (DWI) and (**E**) low signal intensity on the apparent diffusion coefficient (ADC) map.

**Figure 2 medicina-58-00581-f002:**
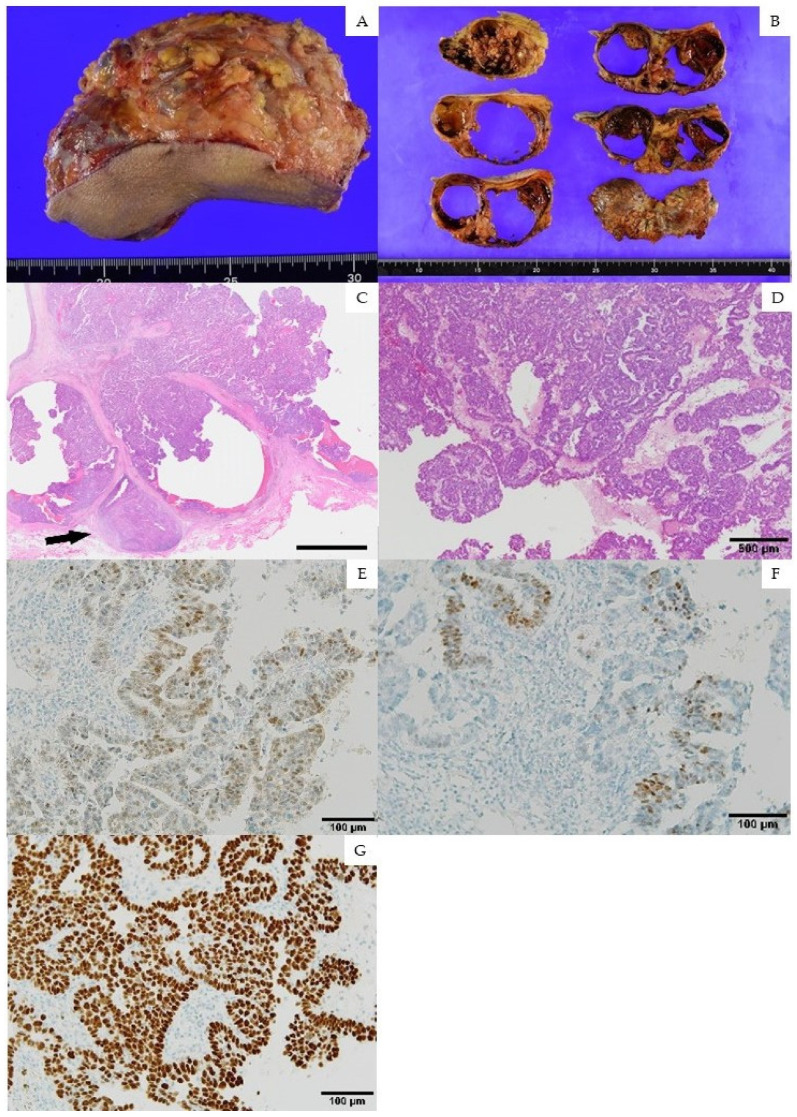
Macroscopic and histopathological findings of the inguinal tumor. (**A**,**B**) Macroscopic images of the inguinal tumor. (**C**–**G**) Photomicrographs of the surgically removed inguinal tumor. (**C**) Multilocular mass encapsulated and separated with fibrous septa, accompanied by a marginal lymph node (an arrow) also involved by metastasis (hematoxylin and eosin (HE) stain, loupe view; bar in the right lower corner, 5 mm). (**D**) The tumor composed of proliferating atypical cells in a well-formed tubulopapillary structure (HE, ×40). (**E**–**G**) The tumor cells showing positivity for PAX-8 (**E**), focally positive for WT-1 (**F**), and overexpression of p53 (**G**) (immunohistochemical stain, ×200).

**Figure 3 medicina-58-00581-f003:**
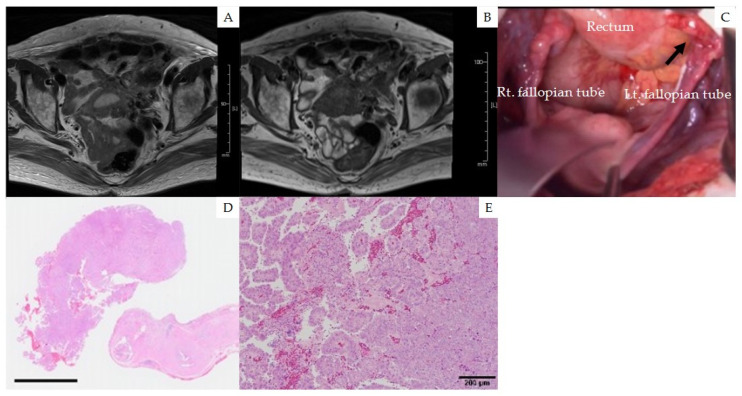
Imaging, surgical, and pathological findings of the fallopian tubes. (**A**) T1-weighted and (**B**) T2-weighted magnetic resonance images showing no apparent disease. (**C**) Intraoperative findings showing mild edema of the left fallopian tube (the black arrow) with no other abnormalities. (**D**, **E**) Photomicrographs of high-grade serous carcinoma arising in the left fallopian tumor. (**D**) The primary tumor defined to the left fimbria with exposure to the surface (HE, loupe view; bar in the left lower corner, 5 mm). (**E**) Papillary adenocarcinoma with a destructive solid pattern (right) indicating stromal invasion (HE, ×100).

**Table 1 medicina-58-00581-t001:** Summary of cases of ovarian or primary fallopian tube carcinoma initially presenting with inguinal lymph node metastasis.

	Year	Age	Origin	Hist	Imaging	Preop	Size1	LN	Size2	Meta	Surg	Rec	PFS	OS
This study	2022	77	FT	HGS	CT, MRI	No	ND	12 cm	2 cm	None	Comp	No	9	9
Bacalbasa [25]	2018	46	Ov	AC	CT, PET	Yes	PET	--	--	PLN	Comp	No	12	12
Metwally [26]	2017	65	Ov	HGS	CT	Yes	--	--	--	None	Comp	No	--	--
Yang [27]	2014	53	Ov	SC	TVUS	Yes	5 cm	3 cm	5 cm	None	Comp	No	60	60
Deka [28]	2013	35	Ov	AC	CT	No	ND	5 cm	ND	None	Comp	No	24	24
Oei [29]	2008	49	Ov	AC	CT	Yes	13 cm	3 cm	13 cm	P, DA	Opt	No	--	--
Ang [30]	2007	59	Ov	SC	CT	Yes	9 cm	3 cm	10 cm	None	Comp	No	7	7
Cho [31]	2006	72	FT	SC	PET-CT	No	ND	2.3 cm	1.2 cm	None	Comp	No	72	72
Manci [32]	2006	58	Ov	LGS	CT, PET	Yes	PET	--	ND	None	Comp	No	19	19
Winte [33]	2001	69	FT	SC	TAUS	No	ND	--	6 cm	#	Opt	Yes	13	20
Cormio [34]	1996	67	FT	SC	CT	No	ND	3 cm	2.6 cm	None	Comp	No	72	72
Kohoe [35]	1992	66	Ov	AC	CT	No	ND	2 cm	15 cm	P, OM	Sub	Yes	8	8
Mcgonigle [36]	1991	59	Ov	EC	CT	Yes	12 cm	4 cm	13 cm	None	Comp	No	32	32
Shulman [37]	1952	63	Ov	AC	AXR	No	ND	1.9 cm	14 cm	None	Comp	No	1.5	1.5

# peritoneum, sigmoid colon, diaphragm, PLN, PAN. FT, fallopian tube; Ov, ovary; Hist, histology; SCC, squamous cell carcinoma; SC, serous carcinoma; AC, adenocarcinoma; HGS, high-grade serous carcinoma; LGS, low-grade serous carcinoma; EC, endometrioid carcinoma; CT, computed tomography; TVUS, transvaginal ultrasonography; TAUS, trans-abdominal ultrasonography; PET, positron emission tomography-computed tomography; AXR, abdominal X-ray; P, peritoneum; DA, diaphragm; PLN, pelvic lymph node; PAN, para-aortic lymph node; OM, omentum; Size1, tumor size at first presentation; Preop, preoperative diagnosis; LN, size of inguinal lymph node metastasis; Size2, tumor size of macroscopic finding; Meta, site of metastasis; Surg, surgery; Rec, recurrence; Comp, complete surgery; Opt, optimal surgery; Sub, suboptimal surgery; PFS, progression-free survival (months); OS, overall survival (months).

## Data Availability

All the studies used in the literature review are published. The data of case presentation are available from the corresponding author upon reasonable request.

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
