# Peer review of "Primary Fallopian Tube Carcinoma Presenting with a Massive Inguinal Tumor: A Case Report and Literature Review"

_medicina, 2022, doi:10.3390/medicina58050581_

Round 1

Reviewer 1 Report

The article is well written, describes a case of a rare cancer type (PFTC) with an unusual presentation (inguinal tumor) and challenging differential diagnosis (unknown origin), and provides recommendations for diagnostic (diagnostic laparotomy or PET-CT) and surgical procedures (resection of the adnexa) of similar cases. Although some case reports have been published in this tumor type, due to the special presentation of the tumor (larger metastatic tumor than the primary tumor) with a high risk of misdiagnosis, it is important to publish such a case and remind the scientific community of this possibility.

The introduction presents adequate literature review about the tumor type, while in the discussion, the authors review literature pertinent to the case in a well-structured format. The case is adequately presented with the figures appropriately supporting the statements of the manuscript. The discussion and conclusion parts emphasize why the case is important to medicine. It is appreciated that the discussion mentions the limitations related to this case report.

It could have made the manuscript and the relevance of the applied targeted treatment more valuable if tumor NGS analysis results were presented, however, homologous recombination repair deficiency and BRCA1/2 status is reported.

Some minor comments:

L122 Please refer to the homologous recombination repair deficiency test method and BRCA1/2 test used, and the samples (tumor/germline?) for these tests. If these analyses were conducted using tumor tissue, please rephrase the sentence in line 122: “The patient had homologous recombination repair deficiency” to “the patient’s tumor showed homologous recombination repair deficiency”.

In Table 1, the patient of this study is indicated to be 70 years old, however, throughout the manuscript the patient was claimed 77 years old. Please, align these occurrences.

L212 and L222: PFTC is misspelled as PTFC

L213: Please, unfold the HGSC abbreviation

L138: “breast carcinoma gene mutations” if you want to refer to BRCA genes, please use standard nomenclature like breast cancer 1 and 2 genes

Author Response

Response to the Reviewers’ comments

We would like to thank the Editor and the Reviewers for the helpful comments. The following are our point-by-point responses to the comments along with descriptions if the revisions made to the manuscript. The revisions made to the manuscript are indicated using the “track changes” function of Microsoft Word.

Reviewer #1

The article is well written, describes a case of a rare cancer type (PFTC) with an unusual presentation (inguinal tumor) and challenging differential diagnosis (unknown origin), and provides recommendations for diagnostic (diagnostic laparotomy or PET-CT) and surgical procedures (resection of the adnexa) of similar cases. Although some case reports have been published in this tumor type, due to the special presentation of the tumor (larger metastatic tumor than the primary tumor) with a high risk of misdiagnosis, it is important to publish such a case and remind the scientific community of this possibility.

The introduction presents adequate literature review about the tumor type, while in the discussion, the authors review literature pertinent to the case in a well-structured format. The case is adequately presented with the figures appropriately supporting the statements of the manuscript. The discussion and conclusion parts emphasize why the case is important to medicine. It is appreciated that the discussion mentions the limitations related to this case report.

It could have made the manuscript and the relevance of the applied targeted treatment more valuable if tumor NGS analysis results were presented, however, homologous recombination repair deficiency and BRCA1/2 status is reported.

Reply:

We appreciate these useful comments from the Reviewer. We have revised the manuscript carefully according to the Reviewer’s comments. We believe the revised manuscript now addresses the Reviewer’s concerns.

Reviewer #1, comment #1

L122 Please refer to the homologous recombination repair deficiency test method and BRCA1/2 test used, and the samples (tumor/germline?) for these tests. If these analyses were conducted using tumor tissue, please rephrase the sentence in line 122: “The patient had homologous recombination repair deficiency” to “the patient’s tumor showed homologous recombination repair deficiency”.

Reply: Line 127

Thank you for your helpful comments. As the Reviewer has pointed out, we have clarified the method of homologous recombination repair deficiency test. We also revised the sentence per the reviewer’s suggestion.

Reviewer #1, comment #2

In Table 1, the patient of this study is indicated to be 70 years old, however, throughout the manuscript the patient was claimed 77 years old. Please, align these occurrences.

Reply: Table 1

According to the reviewer’s comment, we have revised Table 1.

Reviewer #1, comment #3

L212 and L222: PFTC is misspelled as PTFC

Reply: Lines 228, and 239

Thank you for your helpful comment. Per the reviewer’s suggestion, we have revised the sentence.

Reviewer #1, comment #4

L213: Please, unfold the HGSC abbreviation

Reply: Line 229

According to the reviewer’s comment, we have clarified the abbreviation of HGSC.

Reviewer #1, comment #5

L138: “breast carcinoma gene mutations” if you want to refer to BRCA genes, please use standard nomenclature like breast cancer 1 and 2 genes

Reply: Lines 127, and 145

Thank you for your suggestion. We have revised accordingly.

Reviewer 2 Report

Thank you for your well-prepared and fine work. In the article, a case of primary fallopian tube carcinoma and detailed literature review on this case were well presented. Magnetic resonance images, macroscopic and microscopic (H&E and IHC) pictures of the case were been prepared in an easy and understandable format. Detailed information on diagnosis and treatment approaches in similar cases were also given. I believe that this study will make important contributions to future studies on this case presented.

Author Response

Response to the Reviewers’ comments

We would like to thank the Editor and the Reviewers for the helpful comments. The following are our point-by-point responses to the comments along with descriptions if the revisions made to the manuscript. The revisions made to the manuscript are indicated using the “track changes” function of Microsoft Word.

Reviewer #2

Thank you for your well-prepared and fine work. In the article, a case of primary fallopian tube carcinoma and detailed literature review on this case were well presented. Magnetic resonance images, macroscopic and microscopic (H&E and IHC) pictures of the case were been prepared in an easy and understandable format. Detailed information on diagnosis and treatment approaches in similar cases were also given. I believe that this study will make important contributions to future studies on this case presented.

Reply:

We sincerely appreciate the reviewer’s positive comments. We trust that the revised manuscript will now be suitable for publication in Medicina.
